# Where Do We Stand in the Management of Oligometastatic Prostate Cancer? A Comprehensive Review

**DOI:** 10.3390/cancers14082017

**Published:** 2022-04-16

**Authors:** Gómez Rivas Juan, Fernández Hernández Laura, Puente Vázquez Javier, Vidal Casinello Natalia, Galante Romo Mᵃ Isabel, Redondo González Enrique, Senovilla Pérez José Luis, Abad López Pablo, Sanmamed Salgado Noelia, Vives Dilme Roser, Moreno-Sierra Jesús

**Affiliations:** 1Urology Department, Hospital Clínico San Carlos, 28040 Madrid, Spain; laura8635@hotmail.com (F.H.L.); m.isabel.galante@gmail.com (G.R.M.I.); rotundum69@gmail.com (R.G.E.); joseluis.senovilla@salud.madrid.org (S.P.J.L.); pabloabadlopez@gmail.com (A.L.P.); rvivesdilme@gmail.com (V.D.R.); dr_jmoreno@hotmail.com (M.-S.J.); 2Health Research Institute of the Hospital Clínico San Carlos (IdISSC), 28040 Madrid, Spain; javierpuente.hcsc@gmail.com (P.V.J.); nata.vidal@gmail.com (V.C.N.); noelia.sanmamed@salud.madrid.org (S.S.N.); 3Medical Oncology Department, Hospital Clínico San Carlos, 28040 Madrid, Spain; 4Radiation Oncology Department, Hospital Clínico San Carlos, 28040 Madrid, Spain

**Keywords:** prostate cancer, oligometastatic, new imaging techniques, local therapy, metastasis-directed treatment, systemic therapy

## Abstract

**Simple Summary:**

Oligometastatic prostate cancer is an intermediate stage between localised and metastatic disease. Today, there are many advances in the diagnosis of this stage of the disease, with the appearance of new imaging techniques and treatments, thanks to the development of new modalities, both local and systemic therapies, the emergence of personalised medicine, and theragnostics.

**Abstract:**

Oligometastatic prostate cancer (OMPC) is an intermediate state between localised disease and widespread metastases that includes a spectrum of disease biology and clinical behaviours. This narrative review will cover the current OMPC scenario. We conducted comprehensive English language literature research for original and review articles using the Medline database and grey literature through December 2021. OMPC is a unique clinical state with inherently more indolent tumour biology susceptible to multidisciplinary treatment (MDT). With the development of new imaging techniques, patients with OMPC are likely to be identified at an earlier stage, and the paradigm for treatment is shifting towards a more aggressive approach to treating potentially curable patients. Multimodal management is necessary to improve patient outcomes due to the combination of available therapies, such as local therapy of primary tumour, metastasis directed therapy or systemic therapy, to reduce tumour load and prevent further disease progression. Additional prospective data are needed to select patients most likely to benefit from a given therapeutic approach.

## 1. Introduction

Oligometastatic disease is an intermediate state between locoregionally-confined and disseminated malignancy limited in the extent and number (≤5) of metastatic (M1) sites [1]. Hellman and Weichselbaum originally came up with this definition in 1995. Today, the biology of oligometastatic prostate cancer (OMPC) and its prognostic and therapeutic profile are not completely understood. There is no clear consensus on the definition of this stage of the disease [2]. 

Estimates of the prevalence and incidence of this disease setting will be modified with improvements in imaging and treatments for localised disease. In addition, it will be possible to reclassify patients from OMPC on traditional imaging to M1 extended disease [3].

The detection of metastases varies across different imaging modalities. Bone scan and computed tomography (CT) represent the standard staging methods. However, in recent years, with the increase in usage of highly sensitive imaging techniques, such as prostate-specific membrane antigen (PSMA), choline, fluciclovine, sodium fluoride (NaF) positron emission tomography (PET) and whole-body magnetic resonance imaging (wbMRI), patients have been identified at earlier stages of the disease, driving the change in treatment selection [4].

Historically, all patients were treated with androgen deprivation therapy (ADT). However, in the past decade, treatment options have increased significantly. As a result, the curative treatment of OMPC is likely to require an approach from various perspectives based on multimodal management. To date, the primary therapies that have shown evidence are local therapy of the primary tumour, based on surgery o radiotherapy, metastasis directed therapy (MDT) and systemic therapy, alone or in combination, improving patient outcomes [5], in addition to the development of new treatments based on personalised medicine, such as genomics and theragnostics.

This narrative review aims to summarise the current evidence regarding therapeutic options for patients with OMPC.

## 2. Evidence Acquisition

This narrative review will cover the current OMPC scenario. We conducted comprehensive English language literature research for original and review articles using the Medline database and grey literature through December 2021. We searched for the following terms: oligometastatic prostate cancer, treatment, local therapy, systemic therapy, metastasis-directed therapy, new imaging techniques and theragnostics. The combination of terms found 368 related articles; the final number of papers selected for this manuscript was 94. Studies with the highest level of evidence and relevance to the discussed topics were chosen with the consensus of the authors.

## 3. Evidence Synthesis

### 3.1. Prevalence of OMPC

Approximately 10% of the new prostate cancer (PC) cases diagnosed worldwide had M1 disease [6], and globally, from all PC diagnoses, 1/5 patients will reach the M1 stage during the natural history of the disease. Incidence rates of M1 PC have increased slowly during the last 10 years in the United States [7].

In 1988, Soloway et al. observed that patients who had a limited number of lesions on bone scans had improved survival outcomes compared to those who presented with a high volume of disease [8]. This data was corroborated by Ost et al., who found that patients who presented with a single M1 site had significantly improved 5-year survival compared to those who had multiple site M1 disease [9]. Although there has been much epidemiologic data on PC for decades, there is no specific data related to OMPC.

It has been demonstrated that as the number of lymph nodes (LN) and distant metastases increases, the prognosis is worse, and patients with high-volume M1 PC have worse outcomes compared to low-volume M1 PC [10]. At the same time, the development of genomics has allowed us to understand different behaviours within limited M1 and widely disseminated diseases [11]. This is crucial for proposing an aggressive treatment approach in patients with M1 PC [12].

### 3.2. Defining Oligometastatic Disease

At present, no consensus definition exists for OMPC. It is a subgroup of PC patients that may locate an intermediate state between localised disease and metastatic disseminated disease (Figure 1). When cancer is confined to a limited number of sites, it may be curable with MDT. The variability of patients with oligometastatic disease and its different prognostic and therapeutic profiles has not yet allowed a complete understanding of the biology of OMPC [1].

OMPC has been defined based on the number of metastases (typically ≤ 5), while the onset of metastases (synchronous, defined as de novo or within 3 months of primary diagnosis vs metachronous or recurrent metastases) or previous TDA (castration-naive vs castration-resistant) is still a matter of debate [13]. Although primary and recurrent oligometastatic diseases are likely to represent distinct biological states, the effect of this distinction is a fact that should be considered in treatment decision-making. 

The CHAARTED study [14] stratifies patients according to the volume of M1 disease. ≥4 bone metastases, including ≥1 outside vertebral column or spine or visceral metastasis, are defined as high volume, and scenarios that are not high volume are defined as low volume M1 disease. The LATITUDE trial is another study that also classifies patients according to the risk of M1 disease [15]. It is necessary to have ≥2 the following criteria to be classified as high risk: ≥3 bone metastasis, visceral metastasis, or International Society of Urological Pathology (ISUP) ≥grade 4. The rest of the situations are classified as low-risk diseases.

In addition to the volume, Surveillance, Epidemiology and End Results (SEER) seem to find differences in overall survival (OS) depending on the location. For example, Medicare analysis of PC patients found that OS was higher in patients with LN metastases (43 months) than in other sites, such as bone metastases (24 months), visceral metastases (16 months) or bone plus visceral metastases (14 months) [16].

Improvement in detection rates for metastases thanks to the development of new imaging techniques could change the definition of OMPC in the future, as the number of lesions and the lesion size or the standardised uptake value (SUV) could be considered [17].

### 3.3. Imaging Modalities in the Detection of Oligometastatic Disease

Non-invasive radiographic imaging facilitates patient selection for MDT, especially with nuclear medicine and molecular imaging. The detection of metastases is highly related to the imaging technique used. Conventional imaging techniques (CT and bone scan) will detect more metastases in patients classified as oligometastatic; however, patients considered non-metastatic on conventional imaging may have oligometastatic disease [18]. 

CT and bone scans are the most used imaging tools for the diagnosis of M1 disease. The sensibility and specificity of CT are <40% and 98% and of the bone scan are 79% and 82%, respectively [19]. For these reasons, the current evidence suggests that conventional imaging is insufficient to define the oligometastatic state and treatment planning. Functional or molecular imaging has made a significant contribution to clinical decisions, such as PET/CT or MRI, with different radiotracers (Table 1).

PET is a functional imaging technique that evaluates tumour metabolism. The most commonly used radiotracers are choline, 1-amino-3-fluorocyclobutane-1-carboxylic acid (fluciclovine), PSMA, 18-fluorodeoxyglucose and NaF.

Choline is a cell membrane precursor. Increased uptake of this radiotracer has been associated with cell proliferation. A meta-analysis that included 1270 patients from 12 different studies undergoing choline PET/CT showed an overall sensitivity and specificity of around 89% [20]. However, the half-life of choline is short (20 min) and requires a cyclotron. For this reason, logistics has hampered its widespread adoption and use in clinical practice.

Fluciclovine is a synthetic amino acid that has greater absorption in cancer cells due to its higher rates of metabolism, with a half-life of 110 min. In a meta-analysis of 6 studies with 251 patients, sensitivity was around 87%, and specificity was about 66% [21].

The main limitation with these tracers is a lower sensitivity, 7–44% for choline and 21–41% for fluciclovine, for patients with a prostate-specific antigen (PSA) level less than 1 ng/mL [22]; although it is remarkable that most of the patients who have OMPC have higher levels of PSA.

PSMA is a membrane glycoprotein that is low expressed in normal prostate tissue but highly upregulated in more than 80% of M1PC with a half-life of 67 min. PSMA is not fully prostate-specific because it is also expressed in other solid tumours, but it has high specificity for patients with PC [23]. PSMA conjugated to radionuclides has been evaluated for imaging; for example, ^68^Ga-PSMA PET/CT and 18 F-DCFPyL have shown to be the most promising and are approved by the Food and Drug Administration (FDA). Still, there is no consensus as to the optimal PSMA radioligand. In patients with PSA levels lower than 1 ng/mL, sensitivities for detection of primary PC lesions were 70%, pelvis LNs were 61%, and specificities were 84% and 97%, respectively [19]. In patients with biochemical recurrence, a higher pre-PET PSA and shorter PSA doubling time (DT) increase the likelihood of a positive ^68^Ga-PSMA PET. Sensitivity and specificity are both around 86% [24].

18-fluorodeoxyglucose is another radiotracer whose role is debatable because of suboptimal performance characteristics attributed to the biology of PC [25].

Compared with conventional imaging, MRI offers anatomical and functional assessment, with a significant improvement in diagnostic performance. WbMRI has shown promise as a modality in evaluating M1 disease and could be helpful for appropriate patient selection for intense multimodality therapy. In a study of 96 patients with newly diagnosed M1 PC, wbMRI classified 28% of patients with OMPC and 52% with M1 castration-resistant PC (CRPC M1). However, it also has limitations such as time-intensive protocols or higher cost, among others [26].

Even though these new imaging techniques have higher sensitivity and specificity than conventional ones, the clinical benefit of detecting metastases remains unclear [27]. Moreover, their prognosis and management are unknown in patients diagnosed with M1 by more sensitive staging procedures. Therefore, in this group of M1 patients, detectable only with PSMA PET/CT, it is unclear whether they should be managed using MDT or systemic therapies [28]. 

However, the ORIOLE clinical trial, which analyses disease progression at 6 months in men with OMPC (SBRT versus observation), has a secondary endpoint that includes the concordance between PSMA PET/CT and conventional imaging techniques in the detection of M1 disease. Patients in whom appreciable disease was detected by PSMA PET/CT and who received consolidation of all detectable disease found significant progression-free survival (PFS) and distant metastasis-free survival (MFS). These data support the use of molecular imaging in the diagnosis of OMPC in conjunction with MDT [29].

Outcomes from clinical trials showing the results of patients with and without metastases detected by novel imaging tools and MRI are necessary before making a treatment decision based on the results. PSMA PET/CT has a higher accuracy for staging than conventional imaging. Still, no oncological outcomes data exist to inform whether the subsequent management selected by the usage of this test has a beneficial impact on cancer prognosis. Therefore, the European Association of Urology (EAU) endorses PSMA PET/CT only for recurrence, but not as a primary staging tool assessment [30].

**Table 1 cancers-14-02017-t001:** Sensitivity and specificity of imaging techniques in the diagnosis of oligometastatic prostate cancer.

Imaging Techniques	Sensitivity			Specificity			Change in Management
	LN Staging	M Staging	Overall	LN Staging	M Staging	Overall	
CT [19]	38%	38%	38%	98%	98%	98%	NA
Bone scan [19]	NA	79%	79%	NA	82%	82%	NA
Whole-body MRI [26]	41%	85%	60%	92%	85%	95%	NR
Fluciclovine PET [21]	NR	NR	87%	NR	NR	66%	NR
Choline PET [20]	62%	80%	89%	92%	89%	89%	18–48% *
PSMA PET [24]	65%	92%	86%	94%	92%	86%	21–41% *

* Results provided by patients with biochemical recurrence. Abbreviations: CT, computed tomography; MRI, magnetic resonance imaging; PET, positron emission tomography; PSMA, prostate-specific membrane antigen; LN, lymph nodes; M, metastasis; NA, not applicable; NR, not reported.

### 3.4. Treatment

#### 3.4.1. Local Therapy to the Primary Tumour

The different options and findings among the different treatment options are summarised in Table 2. The OS of selected patients improves from cytoreductive surgery in other tumour entities, such as renal cell carcinoma. According to preclinical investigations, there appears to be a similar survival benefit after surgery or radiotherapy (RT) of the primary tumour. In addition, a reduction in angiogenesis has been observed in distant lesions away from the local therapy target. This is known as the abscopal effect [31]. It has also been observed that the primary tumour could induce a metastatic microenvironment [32]. This finding has raised the assumption that, even in M1 disease, primary tumour treatment may stop the progression of those M1 lesions already present. Although these hypotheses have been demonstrated only in retrospective studies, aggressive local therapy in M1 disease could improve cancer-specific survival (CSS) and OS [33].

##### RT of Primary Prostate Tumour + ADT

According to the SEER retrospective study, patients treated with local therapy, radical prostatectomy (RP) or brachytherapy showed improved survival and cancer-specific mortality compared to those who did not receive local treatment [34].

Two recent prospective randomised clinical trials have attempted to answer this question. The first trial, known as HORRAD, randomised (1:1) 432 patients with de novo bone M1 PC to ADT +/− RT to the prostate. Both the prostate and the extraprostatic extension were radiated, but the pelvis LNs were not included. OS, which was the primary endpoint, did not significantly improve with the addition of RT (median 45 months for RT versus 43 months for ADT alone), but time to PSA failure was modestly improved with RT (15 months versus 12 months). However, 63% of patients had >5 bone metastases with a median PSA prior to randomisation of 145 ng/mL, so there was a high percentage of patients with a high M1 burden. Despite that the primary endpoint was not significant, OS in the RT arm for patients with low M1 burden (<5 bone metastases) showed a trend favouring RT. These results suggest that patients with the best prognosis would benefit from local therapy to primary care. This trial was based on the selection of patients considering the bone scan alone, without including visceral disease and patients with high M1 burden were included given the high median PSA of 145 ng/mL [35].

The other large clinical trial is STAMPEDE, a multicentre, multiarm, randomised controlled trial (arm H). It included 2061 patients with de novo M1 PC without prior therapies randomised 1:1 to the standard of care (SOC) (ADT +/− docetaxel) versus SOC plus RT to the primary tumour. Almost 42% of patients in either group had low-burden M1 disease, defined according to the CHAARTED criteria, and only 18% of patients received chemohormonal therapy and RT. The RT arm showed no benefit in terms of OS; however, an improvement was observed in patients with low M1 burden (3-year OS was improved by 8% from 73% to 81% in the RT group for patients with low burden). The group of patients with a high M1 burden did not benefit from RT in terms of OS or PFS [36]. Because SOC included ADT alone in 80% of patients, it is unclear whether the benefits observed with RT persist with systemic combination therapies. 

Despite these limitations in both clinical trials, for patients with newly diagnosed M1 PC with <5 bone metastases (HORRAD) or low volume (STAMPEDE), it is reasonable to offer RT to improved outcomes.

A STOPCAP systematic review and meta-analysis of both studies conclude that treatment with RT of the primary tumour plus ADT is associated with a significant improvement in PFS and provides a 7% improvement in OS in patients with <5 metastases [37].

A large body of literature suggests that RT to the primary tumour for OMPC is a feasible therapeutic option. Its role as a single therapy or in conjunction with systemic or surgical strategies remains to be determined. It is unclear whether the primary function of radiation in OMPC is to delay time to ADT or to consolidate potentially curative multimodal approaches. Several clinical trials are currently ongoing that will attempt to answer these questions.

##### Surgery on Primary Prostate Tumours

The benefit of cytoreductive RP (CRP) in this scenario has not yet been defined. Thus far, non-randomised studies have not shown that surgery in M1 patients is associated with a significant increase in OS. However, these studies provide invaluable oncological benefits. The advantages are unquestionable in terms of local symptom control, local progression of the disease and increased PFS.

Retrospective data have been provided regarding the surgical treatment of the primary tumour. In 2015, a small retrospective case-control study included 23 patients with 3 or fewer bone lesions undergoing CRP compared to 38 patients with M1 PC treated with ADT without local therapy. They showed a benefit in PFS and CSS. The technique is reproducible, with adequate patient selection, and it is associated with a low Clavien–Dindo complication rate and an excellent subsequent functional outcome [38]. 

Another retrospective trial with 113 patients with OMPC confirmed as a strong predictor for biochemical PFS the preoperative PSA less than 1 ng/mL after neoadjuvant ADT and another predictor for OS a PSA-nadir less than 1 ng/mL [39]. Jan et al. analysed 79 men with 5 or fewer bone metastases to receive ADT alone or robot-assisted radical prostatectomy (RARP). Surgically treated patients showed improved PFS and CSS compared to the ADT-alone group [40].

To date, several phase II trials have recently been published. The first is a prospective randomised Chinese phase II trial that reported its first results in 2020. It randomised 200 patients with OMPC with <5 metastases with no visceral to ADT versus ADT + local therapy (RP was preferred, but patients who refused surgery or had a nonresectable tumour, RT was offered). RP was the majority in 88.5% of the patients. Improved radiographic PFS for the group with local therapy was observed after a follow-up of 28 months [41]. However, this follow-up time is still insufficient to consolidate these preliminary results.

Another prospective, randomised, phase II, feasibility clinical trial published is the TRoMbone trial. Fifty men with newly diagnosed OMPC were randomised to RP plus pelvic lymphadenectomy plus SOC (ADT +/− docetaxel) or SOC alone. Patients included had 1 to 3 bone metastases. The primary endpoint was the feasibility of randomising within 12 months, and the secondary outcome was quality of life (QoL). Erectile function was not preserved in any patient; 16.7% of patients remained incontinent six months after surgery. The positive margin rate was 41.7%, Gleason 8–10 was 82.6%, and pT3 disease was 87.5%. Of the patients, 82.6% had a 6-month post-operative PSA <1 ng/mL. Operative times, length of stay, complications, surgical margin rates and early biochemical outcomes were similar to RP series for standard indication. Urinary continence outcomes were also similar to those of standard surgery. The surgical group did not suffer worse QoL compared to the SOC cohort. This trial shows that it is safe and feasible to investigate surgery in this setting [42]. Regarding regional LN dissection, there is a pilot study of multimodal therapy. Patients received at least 6 months of neoadjuvant ADT with CRP plus retroperitoneal lymphadenectomy and RT to the prostate +/− pelvic or para-aortic LNs. Almost all patients (95%) reached an undetectable PSA with a multimodal approach, 25% after ADT, 50% after surgery and 20% after RT. The number of patients who achieved undetectable PSA was higher when adding each element of the multimodal approach. Regarding the follow-up period, the primary endpoint of undetectable PSA after testosterone recovery was achieved in 1/5 of patients [43].

Thus far, CRP in M1 patients has not been contemplated outside of clinical trials and is not included in the Clinical Guidelines. Several clinical trials, such as the Southwest Oncology Group (SWOG) randomised trial phase III, with a planned enrolment of 1273 patients. This trial compares systemic standard therapy with definitive treatment (surgery or RT) versus systemic standard therapy alone [44], G-RAMPP, a similar randomised trial with a planned enrolment of 452 patients, compares best systemic therapy with CRP versus best systemic therapy alone [45] or phase III of TRoMbone, which compares systemic standard therapy with CRP and salvage lymph node dissection (sLND) [46].

These preliminary results suggest that local treatment in patients with OMPC might improve OS. However, until definitive results on cancer outcomes from the above trials become available, the feasibility of CRP can only be hypothesised to be equal to RT as a local treatment option in OMPC.

#### 3.4.2. MDT

MDT therapy is currently a matter of debate. The MDT hypothesis is that the treatment of M1 lesions could prevent the spread of other metastatic lesions and thus improve survival. This could change the paradigm for M1 disease from a palliative to a potentially curable approach in selected patients [47]. Therefore, it includes patients with a low synchronous M1 burden and, more frequently, metachronous deposits after radical therapy, including nodal and bone metastases, to delay the initiation of ADT or prolong the time to progression.

A drawback of MDT approaches is that data on targeting metastases using surgical management has been limited to retroperitoneal LNs as part of sLND. In contrast, stereotactic body radiotherapy (SBRT) has been used for nodal, bone or visceral disease [48].

##### Resection of Distant Metastases

According to the published literature, there is a subgroup of patients who can achieve prolonged clinical recurrence-free survival (RFS) with sLND, with or without ADT.

Considering the largest series of patients undergoing sLND, within one year of follow-up, only 25% of men developed clinical recurrence. A study of sLND informed 8-year clinical RFS of 38% for retroperitoneal nodal disease [49], and another study reported a 5-year clinical RFS of 34% [50]. Moreover, to predict the benefit of sLND in patients with nodal recurrence, it has created a risk stratification tool, which includes the following items: Gleason grade, number and site involvement of nodal metastases, time from RP to biochemical recurrence, ADT at time of imaging techniques and PSA level at sLND [51].

These emerging data suggest that surgery could be a valid therapeutic option in the management of OMPC; even sLND for oligometastatic nodal recurrence after local therapy is associated with good oncologic outcomes but needs to be validated in prospective trials.

##### SBRT

The usefulness of SBRT was validated in the Stereotactic Ablative Radiotherapy for the Comprehensive Treatment of Oligometastases (SABR-COMET) trial. It is a randomised, open-label phase 2 study that included 16% PC patients. After stratifying by the number of metastases (1–3 versus 4–5), patients were randomised 1:2 to receive palliative SOC treatments alone (ADT/chemotherapy) or SOC plus SBRT for all metastatic lesions. SBRT improved PFS and OS in patients with oligometastases when used in addition to SOC systemic therapy [52].

The STOMP trial was a multicentre, randomised, phase II study of recurrence oligometastatic hormone-sensible PC (HSPC) observed in choline PET/CT. Patients were randomised 1:1 to surveillance or MDT of all detected lesions (surgery or SBRT). After a median follow-up of 3 years, MDT was associated with a higher ADT-free survival with a median of 21 months compared with 13 months for surveillance alone [53]. In 2020, STOMP longer-term results were reported, confirming the previous ones; 5-year ADT-free survival was 8% for the surveillance group and 34% for the MDT group [54].

Recently, the results of the ORIOLE trial (randomised phase II) were reported. Fifty-four patients with recurrent oligometastatic HSPC based on conventional imaging received SBRT versus observation. Six months after randomisation, progression was observed in 19% of patients under SBRT vs 61% of the control group. Treatment with SBRT improved median progression-free survival (PFS) (not reached vs 5.8 months) [29].

These last two studies only included patients with ≤3 metastases.

However, it remains uncertain whether MDT improves survival in patients with OMPC despite the benefit of this therapy on progression-free and ADT-free survival. Phase 3 results are expected to reach robust conclusions or confirm phase 2 results.

Other prospective single-arm studies have reported similar outcomes. One of them, known as POPSTAR, is a prospective trial that included castration-sensitive and castrate-resistant oligorecurrent OMPC and treated them with SBRT. The primary endpoint was feasibility, and 97% of patients completed the treatment. Distant PFS at 1 and 2 years was 58% and 39%, and local PFS was 97% and 93%, respectively [55].

Another study that will further clarify the role of MDT in OMPC is arm M of STAMPEDE. This study tests if surgery is similar to RT as local treatment and whether the addition of SBRT to M1 sites further improves OS [56].

A recent meta-analysis of 23 observational studies with SBRT for recurrent metastases concluded that the proportional rates of local control, PFS and androgen deprivation-free survival were 0.976 (95% confidence interval (CI): 0.96–0.98), 0.413 (95% CI: 0.378–0.477), and 20.1 months (95% CI: 14.5–25.6), respectively. The rate of any acute toxicity was 1.3%, and late grade ≥ 2 toxicity was 1.2% [57]. These results show that local control is excellent, with minimal acute or late toxicity and a median duration to initiation of ADT of 20 months.

The first prospective multicentre randomised phase II trial was PEACE VSTORM. In patients with nodal oligorecurrent PC, this study evaluated the potential of combined whole pelvic radiotherapy and MDT compared to MDT alone [58].

Currently, several phase II and phase III studies recruit patients with recurrent OMPC to assess combinations of MDT and systemic therapy options, such as ADT plus abiraterone + apalutamide [59]. POSTCARD will determine the effect of durvalumab in addition to MDT [60] and another one, Stereotactic Body Radiotherapy With or Without darolutamide for OligoRecurrent Prostate Cancer (DART), which will study the combination of darolutamide with SBRT [61]. A phase III trial will test the role of apalutamide alone or in combination with MDT in patients with recurrent HSPC [62].

##### Theragnostics

The concept of theragnostics refers to a combination of therapy and diagnostics, involving the use of radiopharmaceuticals for diagnosis and treatment, allowing the identification of patients who are candidates for treatment and tracking the response using the same or similar radiopharmaceuticals. Treatment involves the use of target-specific ligands that bind to radioactive atoms. Each radioisotope has characteristic physical properties. The ideal theragnostic target is overexpressed in cancer cells and has low or no expression in normal tissues. This allows for a high dose in the tumour cells with low toxicity in normal tissues. A theragnostic pair comprises two agents with the same targeting ligand but different radionuclides. One side of the couple is specific for diagnostic imaging, and the other is specific for treatment purposes [63].

90–95% of PC overexpresses PSMA, a theragnostic therapeutic target. PSMA expression has been correlated with elevated PSA, high Gleason score and early recurrence, which translates into a more aggressive disease [64]. PSMA is a potential target for theragnostic treatment due to the differential expression of PSMA between normal tissues and cancer cells and the ability of PSMA to be internalised upon binding of antibodies or targeted small molecules. Many radionuclides have been used for therapeutic purposes, such as Lutetium-177 (^177^Lu), Iodine-131 and Actinium-225.

The most used therapeutic nuclide is ^177^Lu. The characteristics of this nuclide are its long lifetime and short tissue penetration, which allow efficient delivery of therapeutic radiation to PC lesions. In addition, these therapies are often administered in multiple doses so that subsequent doses can be adjusted according to the absorbed dose to critical structures and tumour tissues [65].

To date, most clinical trials have focused on using PSMA-targeted radiopharmaceuticals for M1 CRPC. However, high PSMA expression has been observed in earlier stages of the disease, such as high-risk CRPC, recurrent and OMPC, making PSMA a valuable option to prevent progression in these patients or allow more extended periods without ADT.

A small pilot study investigated the use of 177-Lu-PSMA-617. It included 10 patients with HSPC with <10 metastases, no curative treatment options (surgery or RT) and significant PSMA uptake in the tumour. It showed promising response results in terms of PSA decline, and all patients had stabilisation of PSA velocity after 2 cycles. PSA decreased in 3 patients, and one of them had a complete biochemical response after 24 weeks [66].

Several ongoing clinical trials have examined the use of ^177^-Lu-PSMA-617 in OMPC. A phase II clinical trial is currently recruiting patients [67]. A clinical trial, called UpFrontPSMA, conducted in men with a new diagnostic of high-volume OMPC, investigates the use of ^177^-Lu-PSMA-617 in combination with docetaxel compared to docetaxel alone [68]. Another study, a phase III trial, compares ^177^-Lu-PSMA-617 in combination with SOC (ADT and androgen receptor directed therapy) with SOC alone.

The results of these trials can provide us with much information on the use of these radiopharmaceuticals in combination in the management of patients with OMPC [69].

At present, theragnostic in the treatment of OMPC has not yet been considered out of clinical trials.

#### 3.4.3. Systemic Therapy

##### Chemotherapy Agents (Docetaxel)

ADT has been the only evidence-based treatment option for prolonging outcomes for newly diagnosed M1 PC patients for decades. Various studies tested new treatments: chemotherapy agents or new androgen receptor-targeted agents (ARTA). Today, systemic therapy with ADT alone or combined with other agents is the SOC for patients with HSPC.

The randomised phase III trials CHAARTED, GETUG-AFU and STAMPEDE (arm C) investigated the effect of adding docetaxel to ADT in the treatment of M1 HSPC.

Of note, most of the studies on systemic treatments are designed for all patients meeting the criteria of M1 HSPC, and low volume/low-risk populations (de novo or metachronous) could meet the definition of oligometastatic patients. However, the sub-analysis of OMPC done within this setting has methodological drawbacks, as it was not an objective of the studies.

The CHAARTED results revealed significantly longer OS than treatment with ADT alone. The median OS was 13.6 months longer with ADT plus docetaxel than with ADT alone. However, this survival benefit was only significantly achieved in high-volume patients after a median follow-up of 53.7 months. No OS benefit was demonstrated in low-volume disease [70].

Subsequently, results from the multi-arm STAMPEDE trial were published, showing a survival benefit in the M1 subgroup of arm C, in which docetaxel was added to SOC. In addition, patients had better survival after the addition of docetaxel, after a median follow-up of 78.2 months [71].

However, the results of the GETUG-AFU 15 trial were published in 2013 and showed no survival benefit from the addition of docetaxel to ADT. Of the patients who received docetaxel, 32% developed metastases. The definition of high or low volume disease was based on stratification of the CHAARTED trial. A nonsignificant 20% reduction in the risk of death in the high-volume group was reported by adding docetaxel after a median follow-up of 83.9 months. No survival improvement was observed in the low-volume subgroup [72].

It is difficult to know why patients with low volume disease HSPC had a benefit from adding docetaxel to ADT in the STAMPEDE trial but not in CHAARTED and GETUG-AFU studies. However, patients with low-volume metachronous HSPC have favourable outcomes compared with low-volume synchronous HSPC [11]. This leads to fewer events occurring, and no statistical difference in the outcome might be seen. This could be because most patients in the STAMPEDE trial had synchronous M1 disease, and in CHAARTED and GETUF-AFU, metachronous OMPC were approximately 50% and 30%, respectively.

Based on these data, it can be concluded that docetaxel with ADT should be considered a possible treatment option in patients with synchronous HSPC. However, this is unclear in patients with metachronous HSPC [55].

A meta-analysis, which included four studies using docetaxel in this setting, demonstrated a 9% absolute improvement in 4-year survival [73].

##### New Androgen Receptor-Targeted Agents

The LATITUDE [74] and STAMPEDE (arm G) [75] trials are the higher evidence trials supporting the use of abiraterone plus ADT in patients undergoing multimodal therapy.

The primary endpoint of both trials was OS and showed a significant OS benefit in both. In LATITUDE, the hazard ratio (HR) in high-risk M1 patients was 0.62. In STAMPEDE, the HR in the overall population (M1 and non-M1) and in the subgroup of M1 patients was 0.63 and 0.61, respectively. The LATITUDE trial included only high-risk patients. However, a post-hoc analysis from STAMPEDE showed the same benefit regardless of risk or volume stratification. The main secondary objectives were PFS, time to radiographic progression, time to pain, and time to chemotherapy. All of them were in favour of combination therapy. No difference in treatment-related deaths was observed with the combination of ADT plus abiraterone compared to ADT alone. However, in STAMPEDE, 20% of patients discontinued treatment due to adverse effects in the combination arms, compared with 12% in the LATITUDE trial.

A meta-analysis of LATITUDE and STAMPEDE trials showed a 38% reduction in the risk of death with abiraterone plus ADT compared with ADT alone. In addition, an absolute improvement of 14% in 3-year OS and a 28% improvement in 3-year clinical/radiographic PFS compared to ADT alone were observed [76].

In summary, abiraterone acetate plus prednisone combined with ADT should be considered as SOC in patients with de novo metastases. In contrast, in patients with metachronous HSPC, it is unclear whether they would benefit from the addition of abiraterone.

The ARCHES study included patients diagnosed with M1 HSPC who were randomised to receive treatment with ADT plus enzalutamide or ADT plus a placebo. The percentage of patients with treatment for the primary tumour was 26%, while the rest were M1 debut. The primary endpoint of the study was radiographic PFS, and the secondary objectives were OS, time to treatment with a new antineoplastic agent, time to PSA progression, the percentage of patients with undetectable PSA, the rate of patients with an objective response to treatment and time to deterioration of urinary symptoms. The patients were stratified according to tumour volume (according to the criteria defined in the CHAARTED study) and prior treatment with docetaxel. Treatment with ADT plus enzalutamide reduced the relative risk of radiological PFS by 61%. This benefit in PFS was observed in all predefined subgroups. Concerning the secondary endpoints, the time to PSA progression time, time to second treatment, % PSA response and % objective response favoured the ADT plus enzalutamide group [77].

The ENZAMET study included patients diagnosed with M1 HSPC who were randomised to receive treatment with ADT plus enzalutamide or ADT plus a non-steroidal antiandrogen. The percentage of de novo patients was 58%. The primary endpoint of the study was OS, and the secondary endpoints were PSA PFS and clinical PFS. The randomisation was stratified according to tumour volume (with CHAARTED criteria), prior docetaxel scheduling, anticipated anti-resorptive bone therapy, geographic region and co-existing conditions. Treatment with ADT plus enzalutamide reduced the relative risk of death by 33% in the overall study patients. The OS results were unaffected after adjusting for geographic region, disease volume, prior to treatment with docetaxel, antiresorptive therapy and co-existing conditions. Regarding secondary endpoints, both PSA and clinical PFS were higher in the ADT plus enzalutamide group than in the control group [78].

The TITAN study included patients diagnosed with M1 HSPC who were randomised to treatment with ADT plus apalutamide or ADT plus a placebo. The primary endpoints were OS and radiological PFS, and secondary objectives were time to initiation of chemotherapy, time to worsening pain, time to initiation of chronic opioid therapy and time to the occurrence of a skeletal-related event (SRE). For subgroup studies, they stratified patients according to tumour volume (defined in the CHAARTED study) and prior treatment with docetaxel. With a follow-up time of 22.7 months, the 2-year OS was 82.4% in the apalutamide group and 73.5% in the control group. Treatment with ADT plus apalutamide reduced the relative risk of death by 33%. These results on OS were maintained when we compared patients with high and low tumour volumes. The 2-year radiological PFS was 68% in the ADT plus apalutamide group and 47.5% in the control group. Treatment with ADT plus apalutamide reduced the relative risk of radiological progression by 52%. These results for PFS were maintained in all stratified subgroups. In relation to secondary endpoints, superiority was observed for treatment with ADT plus apalutamide in the time to initiation of chemotherapy [79].

These three studies allowed for the inclusion of patients with previous treatment with docetaxel.

Recently, at the American Congress of Medical Oncology focusing on genitourinary tumours (ASCO), new dates were presented from the final analysis of the TITAN study. It was found that after a median follow-up of 44 months, there was a reduction in the risk of death by 35%. In addition, 39% of patients were described to crossover from the placebo arm to the apalutamide arm after the cecum had been opened after the interim analysis results were known. When analysing the OS result adjusting for the crossover patients, the benefit of apalutamide is increased, thus demonstrating a 48% benefit in reducing the risk of death [80].

The benefit of enzalutamide and apalutamide is ultimately independent of tumour volume and whether the PC is de novo or recurrent, confirming the value of ARTA across the full spectrum of patients with OMPC.

A meta-analysis was published in the context of M1 HSPC. It included seven trials with over 7000 patients and compared six therapeutic alternatives in terms of OS, radiological PFS and adverse events. In this analysis, abiraterone and apalutamide were the other options that offered the most significant benefits in terms of OS. Docetaxel also improved OS but substantially increased the risk of adverse events [81]. Similarly, a systematic review of the literature together with a network meta-analysis in which it was observed that patients treated with a hormonal agent (abiraterone, apalutamide or enzalutamide) in M1 HSPC would have a longer OS than those treated with chemotherapy [82].

In 2020, a systematic review and network meta-analysis were published to assess the efficacy of different combinations associated with ADT, and there were no differences in OS between the other alternatives, although the final analyses of several studies were not included [83].

##### Combinations

A combination of local therapy and additional systemic therapy, in addition to ADT, might further prolong disease survival, but to date, there is no evidence. This question remains unanswered until ongoing trials that evaluate combination therapies in patients with M1 HSPC are reported.

A randomised phase III trial called ARASENS studies the combination of darolutamide with docetaxel in patients with M1 HSPC. Data reported for the primary analysis showed a risk of death significantly lower in the darolutamide group (32.5%) than in the placebo group (HR 0.68; 95% CI, 0.57–0.80; *p* < 0.001). Both groups had similar adverse events, and the most common incidences were occurred in about 10% of patients. In addition, both groups had the highest adverse events during the overlapping docetaxel treatment period [84].

Another four-arm randomised phase III trial testing a combination of docetaxel and abiraterone in patients with M1 HSPC is known as PEACE1. Patients were randomised 1:1:1:1 to SOC (continuous ADT or bilateral orchiectomy, with or without docetaxel), SOC plus abiraterone, SOC plus RT to the prostate and SOC plus abiraterone plus RT. A statistically significant improvement in radiographic PFS was observed in the ADT +/− docetaxel +/− RT + abiraterone arm relative to SOC plus RT without abiraterone arm. Radiographic PFS improved from a median of 2.2 years to 4.5 years (HR 0.54; 95% CI, 0.46–0.64, *p* < 0.0001). A secondary endpoint was CRPC-free survival; the addition of abiraterone conferred an absolute benefit of roughly two years to both groups (ADT +/− docetaxel +/− RT and ADT + docetaxel +/− RT). The HR for progression to CRPC was 0.40 [85].

Another one is the STAMPEDE (arm J) trial testing a combination of abiraterone, enzalutamide and ADT in patients with M1 HSPC [86].

##### Genomics and New Agents

PC has a strong genetic component. Information about inherited (germline) or tumour-acquired (somatic) mutations is growing at an unstoppable rate. Although we are still far from the personalised tailoring of therapies according to genetic alterations, we have made considerable progress in recent years. Recent contributions have demonstrated the usefulness of target therapies directed at specific genetic alterations.

The main genetic alterations described in advanced and M1 are mutations in DNA repair genes, mutations in mismatch repair deficiency (MMRd)/microsatellite instability (MSI-H) and those of PTEN loss in the AKT-Pi3k pathway [87].

Some studies show a significant rate of mutations in DNA repair genes, including BRCA2, BRCA1, ATM, CHECK2 and PALB2. Germline mutations in DNA repair genes are present in 8–12% of patients with M1 PC. In a retrospective study, the most frequently mutated germline genes were BRCA2 (5.3%), CHECK2 (2%), ATM (1.6%) and BRCA1 (0.9%). This prevalence is notably higher than that detected in localised cancer (5%) or the general population (3%) [88]. Mutations in the somatic lineage have been described in up to 23% of patients with M1 tumours [89].

There are currently three FDA-approved drugs for the treatment of M1 CRPC with mutations in DNA repair genes: olaparib (PROFOUND Study), rucaparib (TRITON 2 study) and niraparib (GALAHAD study), all of them with significant results in terms of survival or delay in radiological PFS [90]. Other studies are being carried out, specifically talazoparib, associated or not with enzalutamide in M1 CRPC, in the TALAPRO trial [91].

Other known mutations, with lower incidence but with therapeutic impact, include modifications in MMRd or MSI-H such as MLH1, MSH2 and MSH6, PMS2, which are less frequent (5–7% of patients). Responses to anti-PD-1/PDL-1 treatment have been determined, with pembrolizumab receiving the first FDA-agnostic approval for tumours with MSI-H in patients with M1 CRPC [92].

The third prognostic and therapeutic interest pathway is the loss of PTEN, a critical suppressor gene whose loss induces hyperactivation of the PI3K-AKT-mTOR pathway, which is key in survival, growth, proliferation and tumour angiogenesis. This alteration, which appears in 40–60% of patients with CRPC, is estimated to be present in 40% of HSPC [93]. Trials are underway with potent inhibitors of the different molecular forms of AKT, in particular capivasertib, with or without abiraterone in HSPC (CAPItello trial) [94].

**Table 2 cancers-14-02017-t002:** Main studies in the treatment of oligometastatic prostate cancer.

Treatments		Studies	Outcomes
Local therapy	RT	HORRADS (RCT) [34]	OS: HR 1.11; 95%CI, 0.87–1.43; *p* = 4. * OS < 5 lesions (HR, 0.68; 95%CI, 0.42–1.10), 5 to 15 (HR 1.18; 95%CI, 0.74–1.89) and >15 (HR 0.93; 95%CI, 0.66–1.32).
		STAMPEDE (arm M) (RCT) [35]	OS: HR 0.92; 95%CI, 0.80–1.06; *p* = 0.266. PFS: HR 0.76; 95%CI, 0.68–0.84; *p* < 0.0001. * PFS low metastatic burden: HR 0.59; 95%CI, 0.49–0.72; *p* = 0.0001. * 3-year OS low metastatic burden: (81% vs. 73%; HR 0.68; 95%CI, 0.52–0.90; *p* = 0.007).
	Surgery of primary tumour	Culp et al. (SEER datebase) [33]	OS: 67 vs 22.5% *p* < 0.001. CSS: 75 vs 48% *p* < 0.001.
		Heidenreich et al. (retrospective case-control) [37]	CSS: 96% vs. 84%, median of 34.5 months.
		Heidenreich et al. (retrospective cohort) [38]	OS: 85% with RP + ADT.
		TRoMbone (RCT) [41]	Feasibility to randomise: demonstrated. QoL: 0% erectile function, 16.7% incontinent six months after surgery, 41.7% positive margin rate, 82.6% Gleason 8–10, 87.5% pT3.
MDT	Surgery (sLND)	Suardi et al. (prospective study) [48]	8-year clinical recurrence-free survival: 38%
		Rigatti et al. (prospective study) [49]	5-year clinical recurrence-free survival: 34%
	SBRT	STOMP (RCT) [51,52]	ADT-free survival: HR 0.60; 80%CI, 0.40–0.90; *p* = 0.11. Five-year ADT-free survival: 8% surveillance vs. 34% MDT (HR: 0.57; 80%CI, 0.38–0.84; *p* = 0.06).
		ORIOLE (RCT) [53]	PFS: HR: 0.30; 0.11–0.81; *p* = 0.002.
		POPSTAR (RCT) [54]	Feasibility rate: 97%; 95%CI, 84–100%. ADT free 24 months: 48%; 95%CI, 31–75%.
	Theragnostics	Privé et al. (Pilot study) [63]	Stabilisation of PSA velocity: 10/10 PSA decline > 50%: 5/10 PSA decline after 24 weeks: 3/10 Biochemical complete response: 1/10
Systemic therapy	Chemotherapy (Docetaxel)	GETUG-AFU 15 (RCT) [69]	OS: HR 1.01; 95%CI, 0.75–1.36.
		CHAARTED (RCT) [67]	OS: HR 0.72; 95%CI, 0.59–0.89; *p* < 0.001. * OS HV: HR 0.63; 95%CI, 0.50–0.79. * OS LV: HR 1.04; 95%CI, 0.70–1.55.
		STAMPEDE (arm C) (RCT) [68]	OS: HR 0.81; 95%CI, 0.69–0.95; *p* = 0.009.
	Abiraterone	LATITUDE (RCT) [71]	OS: HR 0.62; 95%CI, 0.56–0.78; *p* < 0.001. rPFS: HR 0.47; 95%CI, 0.39–0.55.
		STAMPEDE (arm G) (RCT) [72]	OS: HR 0.63; 95%CI, 0.52–0.76. * OS HR M1: HR 0.54; 95%CI, 0.43–0.69. * OS LR M1: HR 0.55; 95%CI, 0.41–0.76.
	Enzalutamide	ARCHES (RCT) [74]	OS: HR 0.81; 95%CI, 0.53–1.25.
		ENZAMET (RCT) [75]	OS: HR 0.67; 95%CI, 0.52–0.86. * OS HV: HR 0.65; 95%CI, 0.42–0.99. * OS LV: HR 0.38; 95%CI, 0.21–0.69.
	Apalutamide	TITAN (RCT) [76,77]	OS: HR 0.65; 95%CI, 0.53–0.79. OS adjusted by crossover: 0.52. * OS HV: HR 0.70; 95%CI, 0.56–0.88. * OS LV: HR 0.52; 95%CI, 0.35–0.79. rPFS: HR 0.48; 95%CI, 0.39–0.60.

Abbreviations: RT; radiotherapy, RCT; randomised controlled trial, OS: overall survival, CI; confidence interval, HR; hazard ratio, rPFS; radiological progression-free survival, CSS; cause-specific survival, RP; radical prostatectomy, ADT; androgen deprivation therapy, QoL; quality of life, vs; versus, sLND; salvage lymph node dissection, SBRT; stereotactic body radiotherapy, HV; high volume, LV; low volume, HR; high risk, LR; low risk.

## 4. Conclusions

The complexity of the management of these patients and the different adverse effects produced by the variety of treatments must be managed jointly by a multidisciplinary team. Oligometastatic disease is a new clinical state that remains relatively poorly understood. The optimal diagnostic and management of patients with OMPC are changing thanks to the development of new imaging techniques and the emergence of new therapies. New therapeutic options, such as local therapy (RT or surgery of the primary tumour), have been demonstrated to improve outcomes in these patients, as well as systemic therapy with chemotherapy (docetaxel) or ARTA (abiraterone, enzalutamide, apalutamide) in combination with ADT. Moreover, MDT (RT/SBRT o surgery) has been reported as a feasible and safe treatment option and genomics-based therapy and theragnostics. Today, we are witnessing an exciting era of molecular diagnostics intending to achieve precision medicine, in which a multidisciplinary approach to tumours has a crucial role (Table 3).

## Figures and Tables

**Figure 1 cancers-14-02017-f001:**
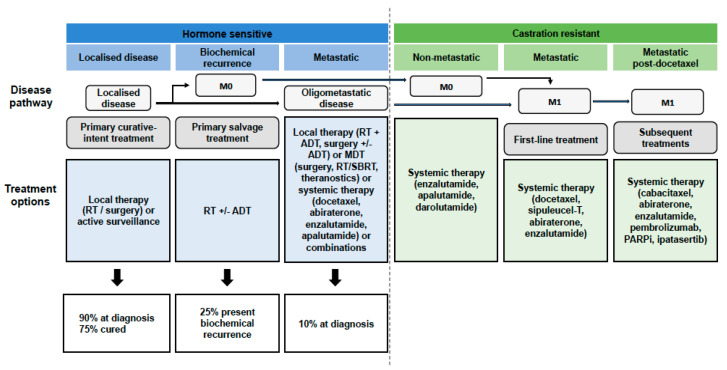
Natural history of oligometastatic prostate cancer. Prostate cancer is split into different clinical states (hormone-sensitive disease shown in blue and castration-resistant disease shown in green) with progressively increasing total tumour burden (represented by the arrows). Below, the different therapeutic options for each of the states are shown. Abbreviations: RT, radiotherapy; M0, non-metastatic disease; ADT, androgen deprivation therapy; MDT, metastasis directed therapy; SBRT, stereotactic body radiotherapy; M1, metastatic disease; PARPi, poly ADP ribose polymerase enzyme inhibitors.

**Table 3 cancers-14-02017-t003:** Key points.

OMPC is defined by the presence of five or fewer metastases on imaging and is a transitional state between localised and M1 disease.
OMPC is a clinical state with inherently more indolent tumour biology susceptible to MDT.
New generation imaging based on PET/CT/MRI scanning has allowed better detection of oligometastatic lesions.
Identifying the 4 clinical scenarios based on risk tumour volume and the diagnosis of de novo or metachronous metastases has been key to guiding treatment.
Local cytoreductive therapies, such as RP with or without pelvis LN dissection and RT, seem to be well tolerated.
MDT (RT/SBRT or surgery) has been reported as a feasible and safe treatment option.
Systemic therapy with chemotherapy (docetaxel) or ARTA (abiraterone, enzalutamide, apalutamide) with ADT has been demonstrated to improve outcomes.
A multimodal approach to patients with OMPC is needed, with evidence of surgery, RT and systemic therapy, alone or in combination, improving patient outcomes.
Further prospective data are needed to best select patients most likely to benefit from a given therapeutic approach.

Abbreviations: OMPC; oligometastatic prostate cancer, MDT; metastatic-directed therapy, PET; positron emission tomography, CT; computed tomography; MRI; magnetic resonance imaging, RT; radiotherapy, LNs; lymph nodes, SBRT; stereotactic body radiotherapy, RP; radical prostatectomy, ARTA; androgen receptor-targeted agents, ADT; androgen deprivation therapy.

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
