# Peer review of "Where Do We Stand in the Management of Oligometastatic Prostate Cancer? A Comprehensive Review"

_cancers, 2022, doi:10.3390/cancers14082017_

Round 1
Reviewer 1 Report
At the beginning, I would like to thank you for the opportunity to review the submitted manuscript, which deals with a truly important but also unclear topic, which is oligometastatic prostate cancer.
The work is very interesting and covers the topic in a comprehensive and up-to-date manner.
The article is well organized and clear
The methods are correctly described and applied.
The final conclusions are adequate to the presented data
References are adequate in number and range.
However, in order for the work to be published, it requires a revisions
Remarks:
- Section - Introduction
- the weakest part of the manuscript - should be thoroughly rebuilt - unfortunately, but numerous repetitions mean that there is no fluidity in the text and the reader loses the right thread - this part may be shorter but must be clear and consistent.
- Section 3.1. Prevalence of OMPC sentences
- paragraph a bit chaotic - no specific data on the presence of OMPC - if not available, please indicate it
- lines 75-76 - Approximately 10% of the newly prostate cancer (PC) cases diagnosed worldwide 75 had M1 disease [6]. 1/5 patients will develop M1 disease. - incomprehensible - 1/5 of which group of patients?
- lines 82-83 - This data was corroborated by Ost el al. in 2016; patients who 81 presented with a single M1 site had significantly improved 5-year survival compared to 82 those who had M1 disease [9]. - this sentence structure blur the differences between the oligometastatic and the M + stage
- lines 89-90 - "Better detection has made possible to reclassify 89 patients with peculiar sensibilities to treatment." - any reference for this statement?
- Section 3.2. Defining oligometastatic disease":
- Line 98: " Thanks to cancer... - wrong term
- Section 3.3. Imaging modalities in the detection of oligometastatic disease
Sentence: "The main limitation with these tracers is a lower sensitivity, 7-44% for choline and 158 21-41% for fluciclovine, for patients with a prostate specific antigen (PSA) level less than 159 1 ng/ml [22]." - it's true but this is general remark for those imaging modalities - patients with OMPC presents with higher levels of PSA at diagnosis.
- Section 3.4.1.2 - Sentence - "With these preliminary results, it appears that local treatment in patients with OMPC improves OS. - On the basis of the presented results, it cannot be said that - it is possible to point to the impact of OS.
- Section 3.4.3.2 - Theragnosis- definitely the more popular and generally accepted term is theragnostics - and should be used. Corrections in all parts of manuscript.
- Section 3.5 - Multidisciplinary approach - This is a review article - the description of the functioning of a multidisciplinary team in a given hospital is not presented in the literature - this paragraph should be deleted. Authors may mention this in the conclusions section.
- Conclusion - change to conclusions
Author Response
Dear Cancers reviewers,
Many thanks for the time spent reading our paper. Your comments will increase the quality of our paper. Please find our comments below:
Reviewer 1:
- Introduction: the weakest part of the manuscript - should be thoroughly rebuilt - unfortunately, but numerous repetitions mean that there is no fluidity in the text and the reader loses the right thread - this part may be shorter but must be clear and consistent.
Replay: introduction has been revised and shortened in order to make it clearer.
- Prevalence of OMPC sentences: paragraph a bit chaotic - no specific data on the presence of OMPC - if not available, please indicate it.
Lines 75-76 - Approximately 10% of the newly prostate cancer (PC) cases diagnosed worldwide 75 had M1 disease [6]. 1/5 patients will develop M1 disease. - incomprehensible - 1/5 of which group of patients?
Lines 82-83 - This data was corroborated by Ost el al. in 2016; patients who presented with a single M1 site had significantly improved 5-year survival compared to those who had M1 disease [9]. - this sentence structure blur the differences between the oligometastatic and the M + stage
Lines 89-90 - "Better detection has made possible to reclassify patients with peculiar sensibilities to treatment." - any reference for this statement?
Replay: paragraph has been rephrased a shortened for better comprehension, and it was indicated that there is no specific data on the presence of OMPC
- Defining oligometastatic disease": Line 98: " Thanks to cancer... - wrong term
Replay: modified.
- Imaging modalities in the detection of oligometastatic disease
Sentence: "The main limitation with these tracers is a lower sensitivity, 7-44% for choline and 21-41% for fluciclovine, for patients with a prostate specific antigen (PSA) level less than 1 ng/ml [22]." - it's true but this is general remark for those imaging modalities - patients with OMPC presents with higher levels of PSA at diagnosis.
Replay: thanks for the remark, a statement of this regard was added.
- Section 3.4.1.2 - Sentence - "With these preliminary results, it appears that local treatment in patients with OMPC improves OS. - On the basis of the presented results, it cannot be said that - it is possible to point to the impact of OS.
Replay: we have modified accordingly.
- Section 3.4.3.2 - Theragnosis- definitely the more popular and generally accepted term is theragnostics - and should be used. Corrections in all parts of manuscript.
Replay: theragnosis was replaced by theragnostic.
- Section 3.5 - Multidisciplinary approach - This is a review article - the description of the functioning of a multidisciplinary team in a given hospital is not presented in the literature - this paragraph should be deleted. Authors may mention this in the conclusions section. Conclusion - change to conclusions
Replay: modified accordingly.
We hope that the manuscript in its current version can be suitable for publication.
With kindest regards,
The authors.
Reviewer 2 Report
In this manuscript authors want to summarize existing data on the management of oligometastatic prostate cancer.
The main drawback is that no formal rules to perform a systematic review have been followed (as PRISMA)
The title should be more formal and declare the article is a review. It should refer to management of oligometastatic prostate cancer, especially to the treatment.
The abstract is not properly structured: methods and results are not presented, and the aim should be highlighted
Authors do not specify if it is a systematic review or a narrative review of literature.
They write “The combination of terms found 368 related articles”, but do not report number of selected papers used for the final analysis.
Moreover, they do not report inclusion and exclusion criteria.
In Conclusion authors write “The new therapeutic options, as local therapy, have demonstrated to improve outcomes in these patients …”, but in the last line declare“insufficient data are available to draw conclusion regarding the new therapeutic options …”. This discrepancy reduces the strength of the collected data.
Conclusions are extremely generic, and authors do not specify what this manuscript adds to current literature.
Author Response
Dear Cancers reviewers,
Many thanks for the time spent reading our paper. Your comments will increase the quality of our paper. Please find our comments below:
Reviewer 2:
- The main drawback is that no formal rules to perform a systematic review have been followed (as PRISMA)
Replay: Yes, indeed is not a systematic review.
- The title should be more formal and declare the article is a review. It should refer to management of oligometastatic prostate cancer, especially to the treatment.
Replay: title has been modified.
- The abstract is not properly structured: methods and results are not presented, and the aim should be highlighted
Replay: abstract has been revised.
- Authors do not specify if it is a systematic review or a narrative review of literature. They write “The combination of terms found 368 related articles”, but do not report number of selected papers used for the final analysis. Moreover, they do not report inclusion and exclusion criteria.
Replay: we have modified materials and methods section accordingly.
- In Conclusion authors write “The new therapeutic options, as local therapy, have demonstrated to improve outcomes in these patients …”, but in the last line declare“insufficient data are available to draw conclusion regarding the new therapeutic options …”. This discrepancy reduces the strength of the collected data. Conclusions are extremely generic, and authors do not specify what this manuscript adds to current literature.
Replay: conclusions have been modified according to the comments by reviewer 1.
We hope that the manuscript in its current version can be suitable for publication.
With kindest regards,
The authors.
Round 2
Reviewer 1 Report
Changes improved quality of the manuscript.
2 remarks left:
- lines 74-75: ... 1/5 patients will develop M1 disease during the natural history of the disease. - Stylistically unacceptable - change to M! stage
- regarding the paragraph on the role of the multidisciplinary team - when writing about the mention in the conclusions, I meant 1 sentence on this issue, not copying the entire paragraph, which, as suggested, should be deleted due to the lack of a significant relationship with the topic of the work
Author Response
Both remarks were corrected in the manuscript. Thanks for your input.